ecology/neuroscience/artificial intelligence

hibernation, *Lasius niger*, insect brain, serotonin, convolutional artificial neuronal network, machine learning

**Author for correspondence:**
Michael Stern
e-mail: michael.stern@tiho-hannover.de

# Seasonal variations of serotonin in the visual system of an ant revealed by immunofluorescence and a machine learning approach

Maximilian F. Bolder[1,2], Klaus Jung[1] and Michael Stern[2]

[1]Institute for Animal Breeding and Genetics, [2]Institute of Physiology and Cell Biology, University of Veterinary Medicine Hannover, Hannover, Germany

MFB, 0000-0002-1743-8688; KJ, 0000-0001-9955-748X; MS, 0000-0002-5088-5439

Hibernation, as an adaptation to seasonal environmental changes in temperate or boreal regions, has profound effects on mammalian brains. Social insects of temperate regions hibernate as well, but despite abundant knowledge on structural and functional plasticity in insect brains, the question of how seasonal activity variations affect insect central nervous systems has not yet been thoroughly addressed. Here, we studied potential variations of serotonin-immunoreactivity in visual information processing centres in the brain of the long-lived ant species *Lasius niger*. Quantitative immunofluorescence analysis revealed stronger serotonergic signals in the lamina and medulla of the optic lobes of wild or active laboratory workers than in hibernating animals. Instead of statistical inference by testing, differentiability of seasonal serotonin-immunoreactivity was confirmed by a machine learning analysis using convolutional artificial neuronal networks (ANNs) with the digital immunofluorescence images as input information. Machine learning models revealed additional differences in the third visual processing centre, the lobula. We further investigated these results by gradient-weighted class activation mapping. We conclude that seasonal activity variations are represented in the ant brain, and that machine learning by ANNs can contribute to the discovery of such variations.

## 1. Introduction

Hibernation, as an adaptation to seasonal environmental changes in temperate or boreal regions, is a feature that reaches back at

least into the age of the dinosaurs [1]. In mammals, hibernation can result in remarkable seasonal changes in brain size [2] and circuitry [3]. Insect brains also display a remarkably high degree of structural plasticity associated with specific behaviours [4,5], but also with age [6,7]. This feature has been studied most intensively in social insects like honeybees and ants [8]. Ants are ecologically very important because they comprise, on average, 15–25% of all terrestrial animal biomass [9]. Ant queens can reach an age up to 28 years or more [10], but ant workers can also live for several years [11]. So far, however, the question of seasonal structural plasticity has not yet been addressed to hymenopteran brains, although seasonal changes can have profound effects on insect physiology and behaviour [12,13]. Changes in the behaviour of workers in ant colonies during hibernation can range from a very immobile state [14] to a similarly active state as during non-hibernation [15] as well as several in-betweens [16]. In addition, egg laying by the queen and brood development usually cease [16]. Compared to other Formicinae species, which usually do not hibernate with brood [17], *Lasius* species hibernate with brood [18]. Behavioural changes during hibernation seem to result from both endogenous processes and exogenous factors [19]. Behavioural plasticity of social insects is largely controlled by neuromodulators, mainly biogenic amines like serotonin, dopamine and octopamine [20,21]. Not surprisingly, division of labour in ants is associated with variations in levels of gene expression [22] and the distribution of biogenic amines [23] in their brains. Expression of these neuromodulators might also differ between the different ant subfamilies, like Ponerinae [24] and Formicinae [25] due to their respective colony structures, with a few dozen to a hundred workers in the former and several hundred thousand in the latter. The general neuroanatomy and chemical architecture of insect brains, including ants, is well understood [26]. Neuromodulatory transmitters, like serotonin, dopamine and octopamine are expressed in a relatively small number of neurons (on average fewer than 100), which, as observed in other insect groups, usually have multimodal sensory input [27] and densely innervate all information processing areas of the central nervous system (CNS). For instance, in the highly repetitive columnar structures of the optic lobe neuropils each column that processes information from a single ommatidium of the compound eye and its surroundings is innervated by serotonergic profiles [28]. This ensures neuromodulatory control of information processing and behaviour, e.g. [29,30]. In the ant brain, distribution and arborization of serotonergic neurons has been described, e.g. by Hoyer *et al*. [24] and Tsuji *et al*. [31]. Serotonin is involved in circadian rhythmicity, and it is important for sleep regulation in both mammals and insects, where serotonin release promotes sleep [32]. Here, we study the distribution of serotonin in the brain of the long-living ant, *Lasius niger* and quantify serotonin immunofluorescence in the optic lobes in comparison between hibernating and active laboratory colonies.

Gross seasonal changes like significant reduction or increase of neuropil volume [33], or dramatic changes in neurotransmitter content would be easily recognized. However, more subtle variations might escape observation, in particular when it is not clear where to expect them, in primary sensory processing areas like optic or antennal lobes, in well-separated central brain regions like the mushroom bodies or central complex, or even in other, far less intensively studied regions of the brain. A promising approach for an unbiased search for seasonal changes could be the use of machine learning by artificial neuronal networks (ANNs) on images of brain sections.

ANNs are currently the most powerful machine learning method for automated image classification [34]. They have already been successfully used to classify two-dimensional images from other arthropods, e.g. Lepidoptera [35] or Araneae [36]. However, approaches so far concentrated rather on species classification [37] than on individual organs. Recently, convolutional ANNs (CNN) have been used on visual data including insects for pest insect monitoring [38], biomass estimation [39] and species identification [40]. Recently, the use of a CNN has been suggested for the detection of cutaneous carcinomas on histological sections [41], or as a new tool for intraoperative tumour tissue visualization during fluorescence-guided brain tumour surgery [42]. Machine learning approaches using random forest algorithms rather than ANNs have been suggested, e.g. for neurotoxicity prediction in human midbrain organoids [43].

In order to improve the performance (e.g. classification accuracy) of ANN models and to reduce the risk of overfitting, original data is often augmented by additional synthetic data [44]. In the case of two-dimensional image data, diverse methods for data augmentation have been proposed, such as geometric transformations, filters, mixing images and feature space augmentation [45]. To increase the interpretability of the results generated by CNN various methods for visualization exist [46]. More recently, gradient-weighted class activation mapping (Grad-CAM) that can be used to generate an explanation from any CNN based network has been proposed [47]. For example, Grad-CAM based approaches have been used to detect COVID 19 cases [48].

Our aim here is not primarily to obtain an ANN model with high classification accuracy, but to use ANNs as a substitute for statistical tests, which would only be applicable to image data after transformation (e.g. into vector or matrix representations) or feature extraction. Such data processing can, however, reduce information about the depicted content of the images. It might also be difficult to estimate the spatial autocorrelation between pixel values when using statistical models for spatial data [49]. Recently, machine learning approaches have been described as more 'tractable' tools for inference with complex data [50]. The idea is that one infers a difference between two sets of images if the machine learning model is able to differentiate the two sets with an accuracy that is better than that of a random classifier. The use of classifier methods for inferential purposes has also been described by Brumback *et al.* [51] Therefore, we here apply a CNN with data augmentation to identify differences in serotonin-immunoreactivity (SI) between hibernating and active ant cohorts.

# 2. Material and methods

## 2.1. Laboratory rearing and hibernation

Three queens of *L. niger* were collected during nuptial flight in July 2019 in Hannover, Germany (52.3759° N, 9.7320° E) and were cultured in test tubes partially filled with water and fitted with a tight cotton plug. The test tubes were stored in the dark at room temperature (21°C). Once nanitics appeared, food in the form of honey and dead fruit flies or parts of juvenile locusts was provided *ad libitum*. In October 2019, the three colonies were first placed in a colder room (10°C) for a week before being transferred to a cool room (4°C) to simulate winter in darkness. During hibernation, workers were observed to be either inactive, forming clusters near to the wet cotton or slightly active on the edge of the colony, similar to the behaviour of *Leptothorax* workers [15]. In March 2020, a total of 46 hibernating workers (cohort H) were removed from the colonies, their head width measured, and their brains processed for immunolabeling as described below. Afterwards, the colonies were brought back to room temperature, stored and fed in the dark as previously described. Being kept and fed in the dark without access to the outside, these ants were considered intranidal workers. A total of 47 active workers (cohort A) were removed from these colonies after 40 days, their head width measured and their brains dissected. Additionally, 35 *L. niger* workers from outside colonies (cohort O) were captured around the Institute of Physiology and Cell Biology, Hannover (52.3759° N, 9.7320° E) in May and processed as described below.

## 2.2. Head width measurements and immunostaining procedure

Ants were cold anaesthetized on ice. Head width (between outer edges of the eyes (figure 1*a*)) was measured with a dissection microscope (Olympus SZ 61), equipped with a digital camera (WUCAM0720PA, OCS.tec, Munich, Germany). A total of $n = 47$ ant brains (cohort A: 15, cohort H: 15, cohort O: 17) were analysed. Brains were dissected in ice-cold phosphate-buffered saline (PBS, 150 mM NaCl, 10 mM sodium phosphate, pH 7.4), immediately fixed in 4% paraformaldehyde in PBS for 1 h at room temperature and washed three times for 5 min in PBS. Brains were stored at 4°C in PBS containing 0.05% sodium azide until all brains of each cohort were dissected and processed side by side as a batch in order to minimize variations. The brains were then embedded in 7% low melting agarose and sectioned into 60 µm frontal sections using a Leica VT1000S vibratome. Sections were permeabilized in 0.3% saponin in PBS with 0.1% Triton x-100 (PBS-T) for 1 h, washed in PBS-T, and blocked in 5% normal goat serum (NGS) in PBS-T for 1 h. Rabbit-anti-serotonin (Sigma S5545, 1 : 2000) and monoclonal mouse-anti-synapsin (SYNORF1, 1 : 50, [52]) in blocking solution were applied overnight. All cohorts were treated with antibodies and blocking serum from the same aliquots. After three washes in PBS-T, secondary antibodies, goat-anti-rabbit Cy3 (Dianova, 1 : 500) and goat-anti-mouse Alexa Fluor 488 (Millipore, 1 : 500) were applied overnight together with 4′6-diamidine-2-phenylidole-dihydrochloride (DAPI, 0.1 µg ml$^{-1}$) as a nuclear counterstain. After three further washes, sections were cleared in 50% glycerol/PBS and mounted in 90% glycerol/PBS. Sections were photographed focusing on the centre of the neuropil using a Zeiss Axioskop equipped with an Axicoam 506 colour camera and Zeiss ZEN 2012 blue edition imaging software, using the same exposure settings for all specimens, with background in the serotonin channel close to zero and maximum intensities at less than 80% of saturation. Two preparations were imaged on a Leica TCS SP5 confocal microscope. For analysis of serotonin-immunofluorescence in the optic lobe, images were processed in ImageJ (http://imagej.nih.gov/ij). To prevent effects due to different sizes of neuropils

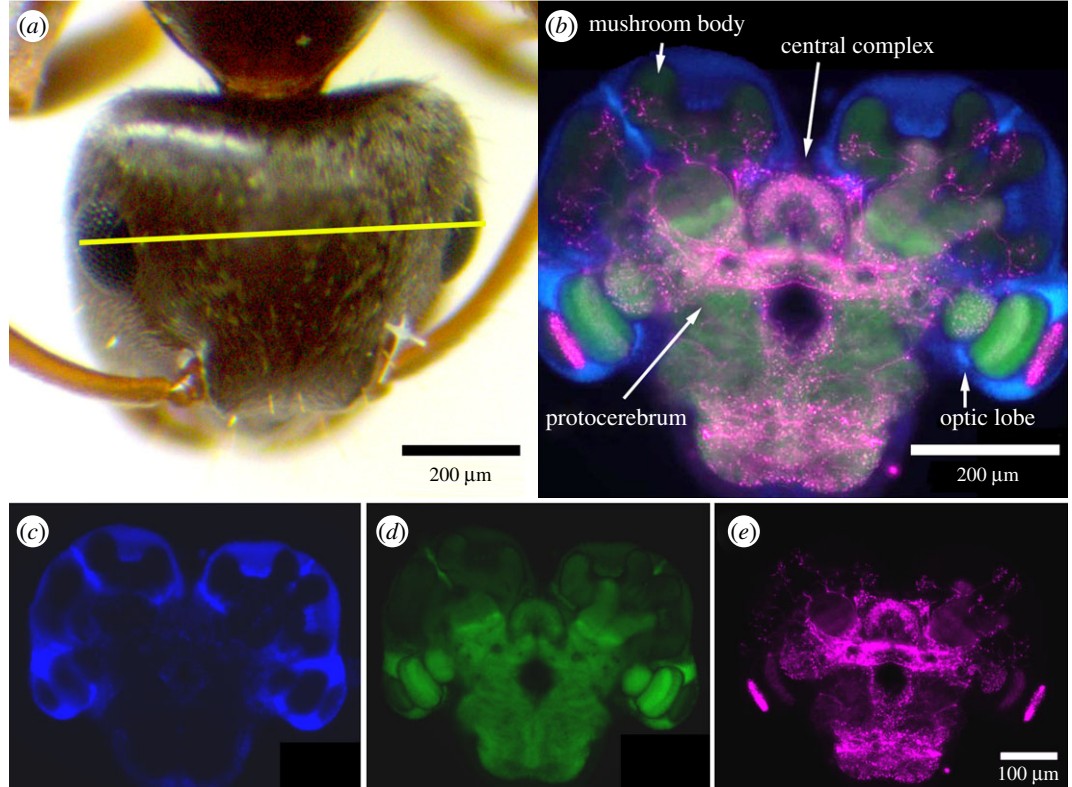

**Figure 1.** *Lasius niger* worker, head and brain section. (*a*) Antero-dorsal view of a head, yellow line indicates head width measurement. (*b*–*e*) Confocal image of a frontal brain section labelled for serotonin (magenta), synapsin (green) and nuclei (blue).

between animals or sections, all images of a given neuropil were cropped at the neuropil borders delimited by the extent of synapsin expression, and reduced to a common resolution (lamina: $100 \times 500$ pixels, medulla: $400 \times 800$ pixels and lobula: $400 \times 250$ pixels). Average relative fluorescent units were measured in ImageJ and plotted in GraphPad Prism 9, and analysed for significant differences by two-way ANOVA and Tukey's *post hoc* test.

## 2.3. Artificial neuronal networks for machine learning

### 2.3.1. Training data

A total of six datasets were created per neuropil containing only the serotonin immunofluorescence images of the respective neuropil. Three datasets, each for the comparison of two cohorts (A versus O, A versus H, H versus O) and three control datasets (A versus A, H versus H, O versus O) consisting of images from only one cohort were generated. Control datasets were randomly split into two arbitrary cohorts and labelled accordingly before being merged again. To avoid a biased split, each dataset was then randomly split between the training set and the validation set 10 times generating 10 randomized versions of each dataset. Dataset size varied depending on the amount of images created per cohort.

### 2.3.2. Data augmentation

Due to the small size of the datasets, there is an increased risk of overfitting the ANNs. Therefore, data augmentation was used to increase the size of the training datasets with additional deformed instances of the original images without affecting the validation set. Data augmentation is a standard technique in fitting ANNs on image data, has been shown to yield models with increased accuracy [53] and is included in the Keras package (https://keras.io/api/preprocessing/image/). Similar to the method used by Zhang *et al.* [54], data augmentation in our approach included rotating the image in a range of 5° and flipping the image horizontally, imitating slight deviation during imaging as well as neuropil orientation in the section.

### 2.3.3. Convolutional neural network structure

The structure of the convolutional neural network was inspired by networks used in recent studies [54–56]. A convolutional layer applied a convolutional filter, also called kernel, over the input image via an element-wise matrix multiplication and the output was mapped to a new matrix and passed through a nonlinear activation function such as 'rectified linear unit' (ReLu) [57]. Pooling layers reduced the in-plane dimensionality of the feature map, via a downsampling operation, which decreased the sensitivity to small distortions [57]. The max-pooling operation was used, which output the maximum value of certain patches. Dropout layers nullified the contribution of some nodes towards the next layer to reduce overfitting [58]. The fully connected layer started with a flattened layer, which turned the final feature maps of the last convolutional layer into a one-dimensional array. This layer was then connected to a dense layer, a dropout layer and another dense layer. The final layer had two outputs and used the softmax activation function [59]. The hyperparameters and structure of the model were determined empirically, via systematically adding or removing layers, and by increasing or decreasing hyperparameters. The datasets were analysed by the model created in R using the Keras [60,61] and Tensorflow [62] package.

### 2.3.4. Model training and evaluation

The number of training epochs was set to 50. Cross entropy was used as a loss function and Adam [63] as the optimizer. Each randomized dataset version was run through the model 10 times (for a total of 100 times for each dataset), with the model being created again after each run. The model's performance was evaluated based on the average validation accuracy achieved on all randomized versions of the respective dataset. Each pairwise dataset performance was compared to the control dataset performance of the cohorts that comprised it (e.g. performance on dataset AO was compared to performance on datasets AA and OO). To rate the differentiability between groups, we use 95% confidence intervals (CIs), and argue that the differentiability is significant if the lower bound of a CI exceeds an accuracy of 50% (random choice).

### 2.3.5. Grad-CAM

To interpret the results of the model on datasets with lobula images, Grad-CAM introduced by Selvaraju *et al.* [47] was used. The previously described model was loaded into a python environment and trained on the dataset of interest. Afterwards, we followed the approach described on the Keras website (https://keras.io/examples/vision/grad_cam/) with slight alterations to generate heat maps from the validation set. These heat maps allowed a first insight into which areas of the images were of interest, which previously could not be detected via immunofluorescence analysis.

# 3. Results and discussion

## 3.1. Head width measurements

*Lasius niger* workers of our laboratory colonies had average head widths of 683 µm (hibernating cohort, $n = 45$), and 687 µm (active cohort, $n = 44$) with very little variation (figure 2a), whereas foraging workers collected outside in summer (outside cohort) were significantly larger (average head width 946 µm, $n = 35$). Older and larger *L. niger* laboratory colonies tend to produce workers of different sizes with a two-peak size distribution [11]. Recent studies show that in *L. niger*, worker size is not directly correlated with internal or external labour [64]. We assume that the large differences in worker size are mainly due to differences in the size and thus the age of the colonies, since in other ant species the size of the colonies correlates with the size of the workers [65]. Since there is a correlation between head and brain size in ants [26], such variations in population size would influence measurements. In our young colonies, we can exclude size variation as a confounding factor. The sizes of the optic lobe neuropils lamina, medulla and lobula did not differ between hibernating and active cohorts, whereas they were significantly larger in the outside cohort as well ($p < 0.0001$, Tukey's *post hoc* test) (figure 2b–d). We detected positive correlations between neuropil size with head width (electronic supplementary material, figure S1).

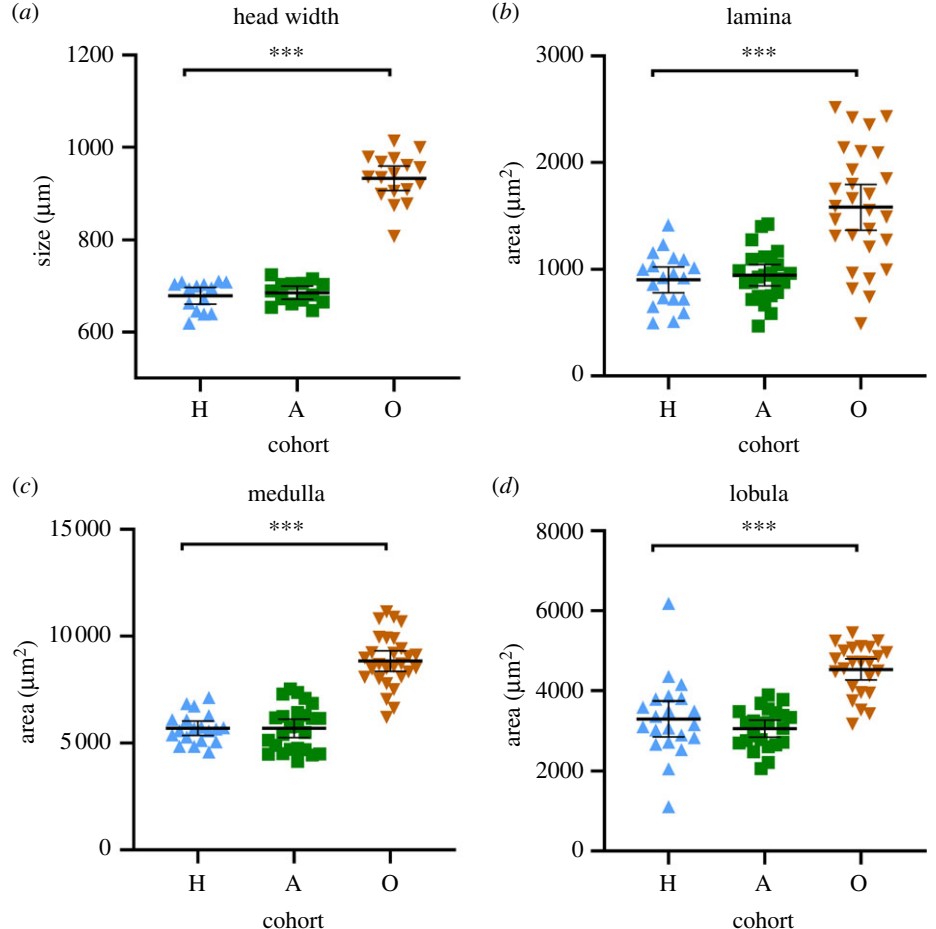

**Figure 2.** Head width and neuropil size of individual workers. Cohorts hibernating (H, blue), active (A, green), outside (O, brown) are indicated on the abscissa. Each point in (a) represents the head width of one worker. Each point in (b)–(d) represents the area of one optic lobe neuropil of one worker on a frontal section. One or two neuropils per worker were measured. Black bars show the mean value, whiskers show the 95% CI. Significant differences between two groups are indicated when present (*** $p < 0.0001$, Tukey's *post hoc* test).

## 3.2. Serotonin-immunoreactivity in the optic lobes

Next, we looked at the distribution of SI in the brain. SI fibres were widely distributed throughout the brain (figure 1c). Serotonergic innervation appears to be particularly strong in the optic lobes, the protocerebrum and the central complex, whereas other areas, such as the antennal lobes and the mushroom bodies appear to receive considerably less serotonergic input. Thus, for our analysis, we focused on the optic lobes (figure 3). In *L. niger* workers collected outside, strong SI was observed in all three optic lobe neuropil areas, lamina, medulla and lobula (figure 3a). Like in other insects [66] including ants [31] the entire lamina was innervated by a dense SI meshwork. In the medulla, SI was largely restricted to the centre, excluding the most distal and proximal layers, similar to the situation observed in the ponerine ant, *Harpegnathos saltator* [24]. In the lobula complex, a more or less homogeneously distributed pattern of SI profiles could be observed. The overall SI pattern in the active cohort of our laboratory colonies (figure 3b) appeared very similar to the outside group. In animals of the hibernating cohort, SI fibres were present in all three optic lobe neuropils as well (figure 3c), but SI appeared to be weaker than in the other cohorts. This impression was substantiated by quantification of SI (figure 4). Average mean immunofluorescence of the lamina was significantly stronger in outside ($p < 0.0001$, Tukey's *post hoc* test) and active ($p < 0.001$) cohorts than in hibernating ants (figure 4a). In the medulla, the outside ants showed slightly but significantly ($p < 0.001$, Tukey's *post hoc* test) higher SI values than both hibernating and active laboratory cohorts did. In the lobula, no significant differences in mean fluorescence intensity could be detected. Measuring average fluorescence intensity over the entire extension of a neuropil area might give reliable results for dense,

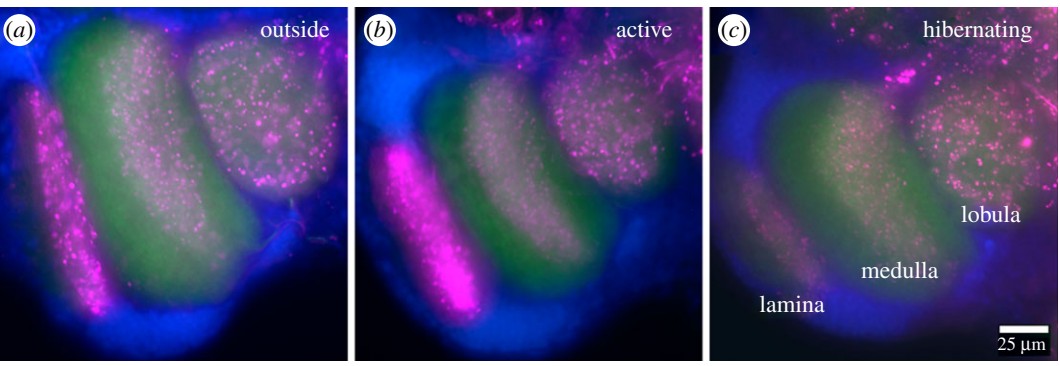

**Figure 3.** Serotonin in the optic lobes of *Lasius niger* workers. Fluorescence micrographs from sections of (*a*) outside, (*b*) active, (*c*) hibernating workers, labelled for serotonin (magenta), synapsin (green), nuclei (blue).

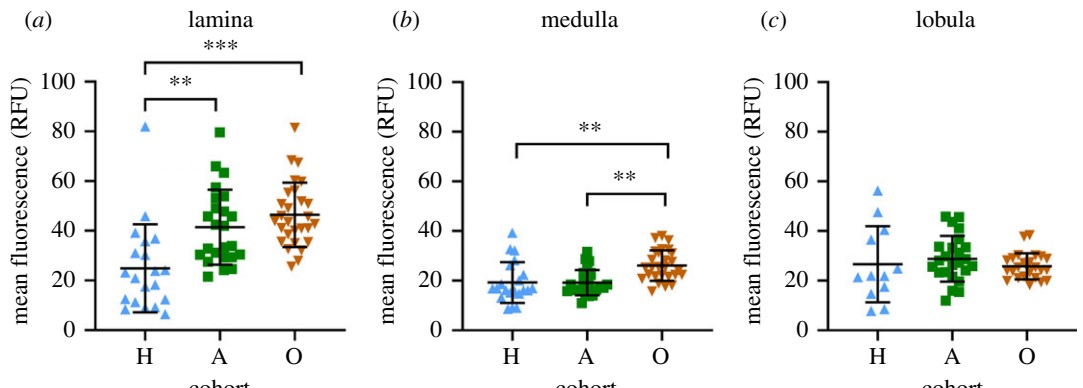

**Figure 4.** Quantification of serotonin immunofluorescence in the optic lobes of *Lasius niger* workers. Cohorts hibernating (H, blue), active (A, green), outside (O, brown) are indicated on the abscissa. Each point represents the average serotonin-immunofluorescence of one optic lobe neuropil of one worker on a frontal section. One or two neuropils per worker were measured. Black bars show the mean value, whiskers show the 95% CI. Significant differences between two groups are indicated when present ($^{**}p < 0.001$, $^{***}p < 0.0001$, Tukey's *post hoc* test).

uniform innervation, but subtle differences in more sparsely innervated areas might escape detection. Other methods, depending on counting the number of immunoreactive profiles per surface area have been used in some studies, e.g. [67], but these depend on threshold effects and bear the danger of observer's bias.

## 3.3. Machine learning approach

We explored whether an objective machine learning approach could help to resolve the problem of detecting differences between SI in the optic lobes of different *L. niger* cohorts. To this end, we divided fluorescent images from each cohort in a random fashion into a training group and a validation group. We trained a CNN with data augmentation in a supervised learning paradigm (figure 5*a,b*), to distinguish between series of fluorescence images from two different cohorts, and as controls, between images from the same cohorts. Figure 5*c* shows an example for data from the laminae of outside and hibernating ants. Whereas the network learned differences in any case during the training phase, correct recognition of a new set of unknown images (validation) only occurred between images from different cohorts (accuracy 84.5%), whereas controls were just randomly assigned with an accuracy of *ca* 50% (figure 5*c*).

We now determined the validation accuracies for all datasets for all three optic lobe neuropils, each pair in 10 independent runs with random splits of the image sets into training and validation groups (figure 6). In all control settings, training and validating images from the same cohorts against each other, validation accuracies varied around 50% (random choice), as expected. In the lamina, the ANN identified both outside and active cohorts as different from hibernating (figure 6*a*), as in our

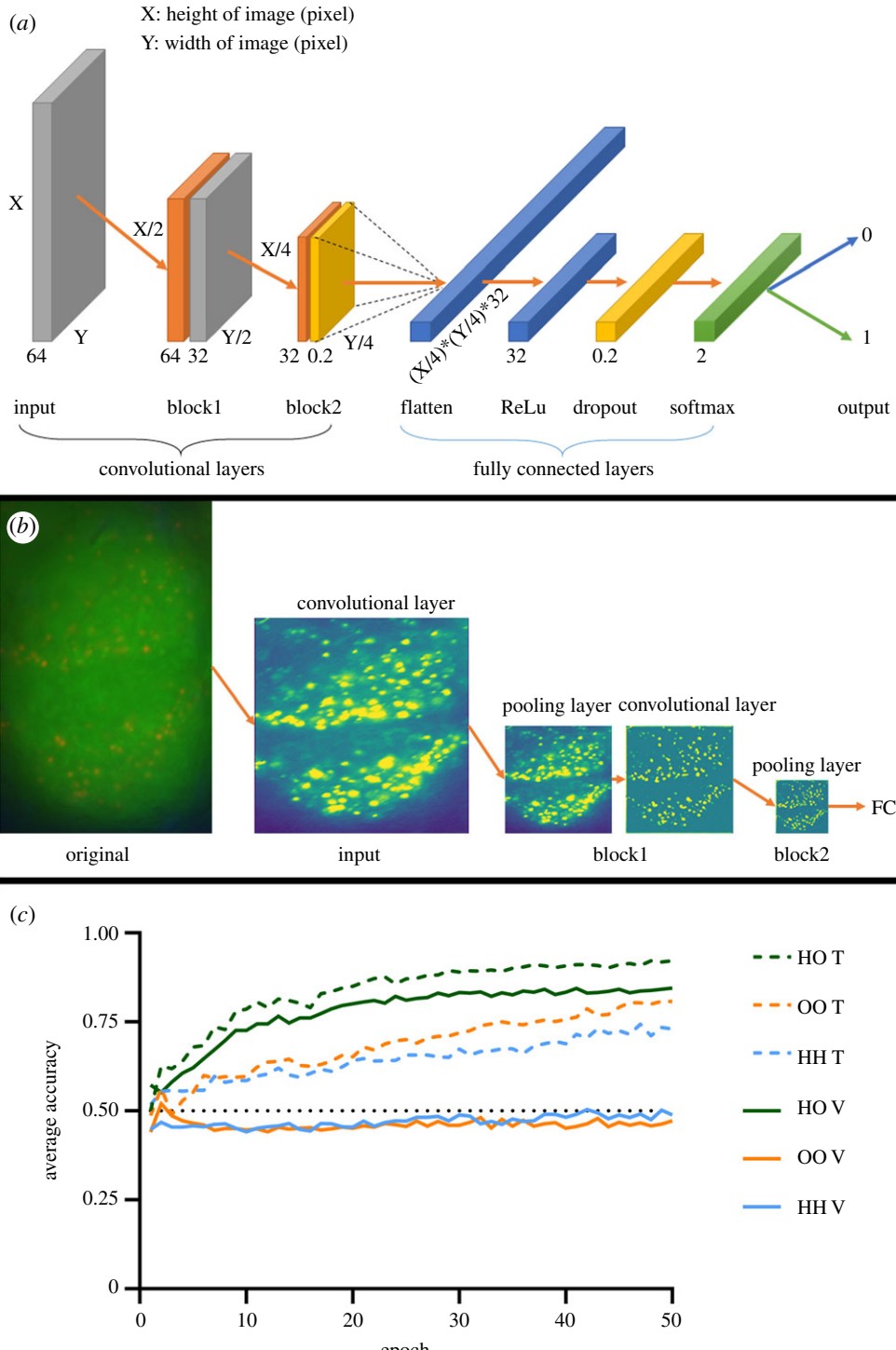

**Figure 5.** Convolutional neural network structure, visualization of the convolutional layers and average performance achieved on the datasets. (*a*) Visual representation of the convolutional network architecture. The shape of the convolutional layers and the flattened layer were dependent on the size of the image used. Grey boxes show convolutional layers, orange boxes pooling layers, yellow boxes dropout layers, blue boxes fully connected layers (with varying functions) and the green box show the last layer, followed by the output. Numbers under convolutional and pooling layers describe the number of filters applied, numbers under dropout layer describe the ratio of dropout, numbers under fully connected and activation layers describe the number of nodes. Flatten, ReLu and softmax are the respective functions associated with the respective layers. All convolutional layers had ReLu as the activation function. Numbers to the left and right describe the shape of the layer, where applicable. (*b*) Example of activations generated per convolutional layer from a lobula image. (*c*) Average accuracy per epoch in the lamina datasets HO, OO and HH, with the average training accuracy per epoch shown as a dashed line and average validation accuracy per epoch shown as a continuous line. The black dotted line indicates an accuracy of 0.5.

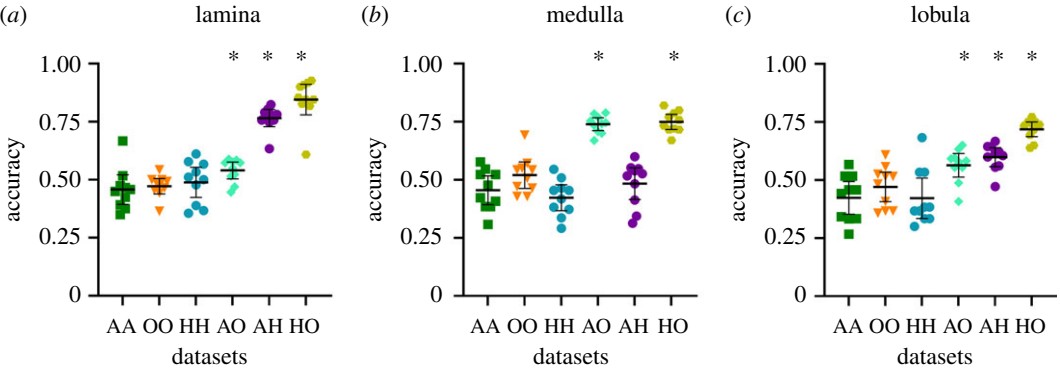

**Figure 6.** Machine learning of differences in serotonin immunofluorescence in the optic lobes of *Lasius niger* worker cohorts. Achieved average validation accuracy of different datasets per split (1–10). The abscissa shows the six datasets, equal letters indicating controls (e.g. AA active compared to active), different letters indicating comparisons (e.g. AH active compared to hibernating). Each point represents the average accuracy of 10 runs on a single dataset randomization after 50 epochs (as in figure 5c). Black bars indicate the mean accuracy of 10 independent randomizations, whiskers indicate the 95% CI, *indicates that the lower bound of the 95% CI is above 50% (i.e. ANN classification is significantly better than random classification).

fluorescence intensity measurements (figure 4a). Likewise, the medullae of outside ants were different from those of both hibernating and active cohorts (figure 6b). However, in contrast to our fluorescence measurements (figure 4c), the ANN discovered significant differences between the lobulae of hibernating *L. niger* and both active and outside cohorts (figure 6c). To interpret the results of the deep learning model on datasets with lobula images, the Grad-CAM introduced by Selvaraju *et al.* [47] was used. It turned out that the heat map of the most relevant learning features coincided well with the distribution of the most prominent SI profiles in the lobula (figure 7).

## 3.4. Significance of serotonin in the insect brain

In the insect optic lobe, visual information is processed by columnar interneurons with increasing receptive field size along the hierarchically ordered neuropil areas lamina, medulla and lobula, in each of which distinct features like colour, shape or movement are processed by local circuits and relayed to deeper brain centres by tangential neurons [68]. These information processing circuits in the insect optic lobe are under neuromodulatory control [69]. 5HT is known to modulate neuronal responses of neurons in all three visual ganglia (lamina, medulla, lobula), but since in most studies serotonin had been bath-applied, the exact site of 5HT action is not clear. Indirect information about the sites of 5HT action can be derived from the distribution of serotonin receptors in all three optic lobe neuropils [70]. In the visual system of many insects, 5HT is involved in circadian rhythmicity, most conspicuously observed in the lamina. There, 5HT is involved in setting the day state [71] and can affect even the physical size of lamina monopolar cells [72]. In *Drosophila melanogaster*, direct modulation of calcium signals in lamina interneurons by serotonin has been demonstrated [70]. We have observed significant differences in SI between laminae of hibernating and active *L. niger* workers. This could point to a more important role of circadian rhythmicity for active than for hibernating colonies. In the lobula, modulation of response characteristics of motion-sensitive interneurons by serotonin has been shown, e.g. in the honeybee [73]. Serotonergic modulation of sensory information processing is not limited to the visual system, but has also been shown for olfaction. For instance, serotonin enhances the sensitivity of the antennal lobe projection neurons in an odour-specific manner [74]. However, we have not observed striking differences in serotonin-immunofluorescence between hibernating and active *L. niger* antennal lobes at a first glance (data not shown). A possible explanation of the stronger effect of hibernation on the visual system than on the olfactory system could be that important carriers of olfactory cues like nest-mates and the queen are still present during hibernation whereas visual input is not. Since the active cohort of our laboratory cultures was reared in the dark, it is remarkable that SI intensities appeared to differ between active and hibernating cohorts both in the lamina and lobula (but not in the medulla). This could hint to a regulation of serotonin synthesis or storage in anticipation of visual input, and as a response to increased temperature and/or availability of food. It would be interesting to test whether changes in serotonin content would follow, precede or simply coincide with seasonal behavioural changes in ants and other insects.

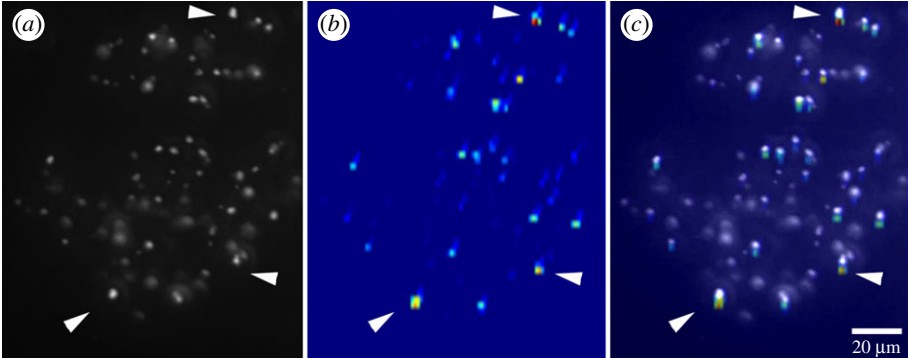

**Figure 7.** Visualization of the class activation map of an outside ant lobula using Grad-Cam on the trained model. (*a*) Original image of serotonin-immunofluorescence as an 8-bit image. (*b*) Class activation map of the image. Bright blue and red spots reflect areas of interest to the model at the time of prediction. (*c*) Overlay of (*a*) and (*b*). Arrowheads indicate three exemplary areas of high interest located in apposition to large diameter neurite profiles.

## 3.5. Methodological aspects

We are aware of the risk that unintended variations of the experimental procedure can add to biological variations as confounding factors when comparing batches of immunolabelled tissue. Therefore, we have attempted to minimize such possible confounding factors wherever we could (e.g. side by side processing of sections in the same solutions, using antibodies from the same frozen aliquot for different ant cohorts). An alternative way to design the study would have been to activate several hibernating colonies, keep the same amount of colonies in hibernation, and later dissect and process them side by side. However, we think that potential variations between (genetically different) colonies would be not less than potential variations between histological processing batches. Furthermore, there were no significant differences in SI between active and hibernating cohorts in the medulla, but only in the lamina and lobula (detected via machine learning), which excludes the possibility of a global, brain-wide variation on SI across seasons in our results. Another possible limitation of our study is the artificial situation in the laboratory, which is also reflected in the size differences to the outgroup of ants collected from the environment. An alternative would be to study wild-caught hibernating and summer ants, instead. However, because of the age- and colony size-related worker size differences, one would need to study the same wild colonies over several years. Furthermore, numerous unobservable confounding factors are likely to influence wild colonies. We think that for a quantitative analysis, a controlled but artificial laboratory environment is more favourable. Fully aware of the fact that the laboratory environment is an artificial situation, we chose outside workers as an outgroup. Interestingly, it turned out that with respect to SI in the lamina, active dark-reared laboratory workers clustered with foraging outside animals rather than with hibernating workers.

## 4. Conclusion

Even with a relatively small sample size, we could detect profound differences in serotonin-immunoreactivity in the visual systems of hibernating and active *L. niger* workers. The strongest differences were found within the most peripheral layer of visual information processing, the lamina, but with the aid of a machine learning approach using a CNN, we could also detect differences in higher-order visual processing centres such as the lobula. We conclude that seasonal behavioural activity patterns appear to be reflected in neurotransmitter content variations in the ant brain, and that the use of machine learning can help to discover subtle differences in insect brain images.

Ethics. This article does not present research with ethical considerations.
Data accessibility. The datasets supporting this article, the ANN code and the GRAD-CAM code with example image set are available in the electronic supplementary material [75] and from the Dryad Digital Repository: https://doi.org/10.5061/dryad.cjsxksn5z [76].
Authors' contributions. M.F.B.: data curation, formal analysis, investigation, methodology, software, visualization, writing—review and editing; K.J.: conceptualization, supervision, writing—review and editing; M.S.: conceptualization, funding acquisition, methodology, project administration, resources, supervision, writing—original draft.
Competing interests. The authors declare that they have no competing interests.

Funding. This publication was supported by Deutsche Forschungsgemeinschaft and University of Veterinary Medicine Hannover, Foundation within the funding programme Open Access Publishing.
Acknowledgements. We thank Gerd Bicker for continuous support and valuable discussions.

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
