## [Peer Review File · Royal Society Open Science]

Review History

RSOS-210932.R0 (Original submission)

Review form: Reviewer 1

Is the manuscript scientifically sound in its present form?

Yes

Are the interpretations and conclusions justified by the results?

Yes

Is the language acceptable?

Yes

Do you have any ethical concerns with this paper?

No

Have you any concerns about statistical analyses in this paper?

No

Recommendation?

Accept with minor revision (please list in comments)

Comments to the Author(s)

In the present manuscript, the authors analyzed the seasonal variations in serotonergic signals in the optic lobes of the ant *Lasius niger*. By combining immunofluorescence, brain imaging, and machine learning, the study shows differences in serotonergic innervation between field-caught and lab-reared individuals and suggests seasonal variation in the ants' visual centers. The originality of the manuscript is the use of Artificial Neural Networks to classify hibernating and active ants. The use of Grad-Cam in this context is innovative and the manuscript is well constructed and experiments are carefully designed. The only major comment for the authors is to consider toning down some statements comparing the "performance" of inferential statistics and machine learning algorithms for detecting differences between treatment groups. These two tools are performing different tasks (making inferences using a random sample of data from a given population, versus algorithms that learn how to assign a class label to examples from the problem domain). The manuscript would benefit from revising statements related to this comparison and, instead, highlighting how machine learning algorithms can be used as an additional tool in neuroethology to complement inferential statistics.

Please find below more detailed comments (page numbers are the numbers provided on the reviewer version of the PDF, they might be off by 1 in the authors' version):

Main comments:

P3 L33-34: Authors state that statistical tests are not applicable to raw image data. I don't think this is accurate as there are examples in the literature where pixel values of pictures are concatenated into vectors and multiple images stacked into matrices for multivariate analysis, analyses of similarity, etc. These tests are however performing a different function: comparing observed distributions to expected ones, while ANN is typically used for classification tasks.

Section 3.2: To correct for shielding effects, average pixel intensities are typically calculated in the same manner for similar volumes (or areas) of both the region of interest (neuropils) and of background signal at the depth of each region of interest. This normalizes for differences between preparations, exposures, etc. Were such steps applied here? If yes, please specify the details in this section.

The authors have employed Tukey's posthoc test to draw comparisons between the three categorical variables. Please clarify if GraphPad Prism 9 corrects for these multiple comparisons by default. If not, employing approaches such as the 'Bonferroni Correction' would be appropriate.

P4 L16 "A few preparations" please specify how many.

Section 3.3.2. Please add references to examples from the literature where this method of data augmentation has been applied. Or to the Keras website given that they describe this in their guidelines.

4.1. If head and brain sizes are correlated, using one of those measures as a covariate in the analysis will help compare differences in the sizes of optic lobe neuropils between active, hibernating, and outside ants while discounting for size-specific effects. Did the variations in sizes of the optic lobe neuropils scale linearly with variations in head or brain size? If not, please clarify why ANOVA was used instead of non-linear models. In addition, have you tried

normalizing by the head width? In doing so, are optic lobe neuropil areas over brain width different between H, A, and O individuals?

4.2. When describing significant differences, please add statistical details in the text as well (unless this is against the editorial guidelines).

4.3. Can you quantify the coincidence of the most relevant learning features and the most prominent SI profiles? (cross-correlation?)

P6 L 16: "However, in contrast to our fluorescence measurements (Fig. 3c), the artificial neuronal network discovered significant differences between the lobulae of hibernating *Lasius niger* and both active and outside cohorts (Fig. 6c)." It seems that Fig 4c would be a better reference in the first half of the sentence. Also, and although micrographs shown in Fig. 3 are most likely randomly selected preparations, they show quite a striking difference between the lobulae of Active (3b) and Hibernating (3c) workers. In this context: 1) Please verify that micrographs are labeled correctly, 2) consider re-ordering as H, A, O, as in the rest of the figures, and 3) are there micrographs that would be representative of the mean serotonergic innervation of each group?

ANN discovered differences between the lobulae of ants across the three categories which the fluorescence measurements did not detect. However, in the first case (ANN), the training of the algorithm benefits from data augmentation, while the inferential statistics might have been limited by the sample size. Could the authors comment if performing a bootstrap on fluorescence measurements would have resulted in a different outcome? Bootstrapping fluorescence intensity measurements could be viewed as analogous to data augmentation in ANN. To clarify, this is not necessarily something that is critical for the manuscript, but because authors put forth the idea of ANN performing "better" at detecting subtle differences, it is important to consider whether the statistics used here could have been limited by the sample size.

Figure 7 / Grad-cam: The immunohistochemistry methods used here label serotonergic neurons. So the micrographs used for training the ANN show the labeled serotonin, synapsin, and nuclei. Is this correct? Or did you train the ANN on images showing only the serotonin immunofluorescence? If the latter, then the overlap between the serotonin image and the class-activation map is expected, right? Because that is the only information available to the ANN during training. Please comment and clarify.

Minor comments:

P2 L32 *Lasius niger* should be italicized.

P3 L9 (and throughout the manuscript) *Lasius niger* should be italicized.

P3 L20 please replace "insects" with "arthropods" because Araneae are not insects but arachnids.

P3 L48 Please consider removing "prematurely", as it is assumed that if killed, it is before the animal's natural death.

P4 L19 Please correct as "neuropil borders and reduced"

P4 L43 Please italicize "*Harpegnathos saltator*"

P6 L27 Please italicize "*Drosophila melanogaster*"

Review form: Reviewer 2

Is the manuscript scientifically sound in its present form?

No

Are the interpretations and conclusions justified by the results?

No

Is the language acceptable?

Yes

Do you have any ethical concerns with this paper?

No

Have you any concerns about statistical analyses in this paper?

No

Recommendation?

Major revision is needed (please make suggestions in comments)

Comments to the Author(s)

In this manuscript, the authors seek to find out if the physiological state of hibernation is represented as changes in serotonin expression in the ant optic lobes. They compare serotonin-immunoreactivity in optic lobes of hibernating versus active laboratory-reared ants as well as wild-caught foragers. Immunoreactivity patterns are compared using traditional statistical analyses as well as machine learning algorithms. The questions considered here are interesting, but the experimental design raises major concerns.

- 1) The primary concern has to do with the handling of the samples used for immunohistochemistry. According to section 3.1 in the Methods, hibernating workers were dissected, and the brains stored for later processing. It is unclear how these brains were stored and for how long. The colony was transferred to room temperature and the next cohort of workers was dissected after 40 days. Were the brains from cohort H in storage that whole time? Were the brains from cohorts A and O also stored for the same amount of time or were all the groups processed at the same time? Many antibodies react very differently in tissue that is stored over a week versus fresh tissue and how that tissue is stored can significantly impact staining. The storage of cohort H brains could account for the decrease in staining. The way this section reads, this experiment was not properly controlled.
- 2) Different batches of stained tissue can also have variations in staining. Here serotonin-immunoreactivity was represented as “average relative fluorescent units” but it’s unclear how this data was normalized to account for variations in global intensity. Also, was the entire 3-dimensional structure of each neuropil included in the analysis even though the brain was sectioned?
- 3) Another concern has to do with rearing the ant colonies in the dark. There is no discussion about the developmental effects of this rearing condition on the ant brain, especially the optic lobes. There are significant differences in brain size and serotonin-immunoreactivity between the wild caught and lab reared ants and it isn’t clear that serotonergic neurons developed as they would in wild animals. Why were the ants not provided with an ecologically relevant light cycle? It would also be more convincing if the authors obtain wild-caught ants during hibernation and compare these to the wild-caught foragers. On the subject of wild-caught ants, the authors mention a bimodal distribution of ant sizes in established colonies. Is this size variation related to division of labor and if so, presumably cohort O were larger because they were foragers. Were ants in the lab reared colonies able to forage and if so, were cohort A

individuals foragers or working within the nest? In other words, were apples compared to apples?

4) The introduction would benefit from more details on what hibernation looks like in this ant species. Are all the ants in a state of torpor or do they move within the colony? Do all ants hibernate at the same time or do some remain somewhat active to perform maintenance tasks? Also more detail on the connection between serotonergic signaling and awake versus sleep states would strengthen the rationale for performing this study.

5) It is difficult to interpret the significance of the artificial neuronal network identifying differences between the lobulae of all the cohorts when traditional statistical methods did not. According to the authors, the most relevant learning features overlapped with “the most prominent SI profiles.” Was the overall fluorescence equal but the distribution different?

6) One minor issue is that scientific names should be italicized in the manuscript.

Decision letter (RSOS-210932.R0)

Dear Dr Stern

The Editors assigned to your paper RSOS-210932 "Seasonal variations of serotonin in the visual system of an ant revealed by immunofluorescence and a machine learning approach" have now received comments from reviewers and would like you to revise the paper in accordance with the reviewer comments and any comments from the Editors. Please note this decision does not guarantee eventual acceptance.

Please submit your revised manuscript and required files (see below) no later than 21 days from today's (ie 14-Jul-2021) date. Note: the ScholarOne system will 'lock' if submission of the revision is attempted 21 or more days after the deadline. If you do not think you will be able to meet this deadline please contact the editorial office immediately.

on behalf of Dr Jake Socha (Associate Editor) and Pete Smith (Subject Editor)
openscience@royalsociety.org

Associate Editor Comments to Author (Dr Jake Socha):

Associate Editor: 1

Comments to the Author:

Both reviewers identified that this study is interesting and provides an original approach. However, there are some major issues that need to be addressed in order for this manuscript to move forward at RSOS. The reviewers have listed numerous concerns. But also please make sure to clarify the main point that machine learning is somehow better than inferential statistics at picking up subtle effects, and address and clarify the potential issue that one treatment group (the hibernation cohort) was kept in a refrigerator for several weeks before immunostaining while the other groups were processed more immediately after dissection.

Reviewer comments to Author:

Reviewer: 1

Comments to the Author(s)

In the present manuscript, the authors analyzed the seasonal variations in serotonergic signals in the optic lobes of the ant *Lasius niger*. By combining immunofluorescence, brain imaging, and machine learning, the study shows differences in serotonergic innervation between field-caught and lab-reared individuals and suggests seasonal variation in the ants' visual centers. The originality of the manuscript is the use of Artificial Neural Networks to classify hibernating and active ants. The use of Grad-Cam in this context is innovative and the manuscript is well constructed and experiments are carefully designed. The only major comment for the authors is to consider toning down some statements comparing the "performance" of inferential statistics and machine learning algorithms for detecting differences between treatment groups. These two tools are performing different tasks (making inferences using a random sample of data from a given population, versus algorithms that learn how to assign a class label to examples from the problem domain). The manuscript would benefit from revising statements related to this comparison and, instead, highlighting how machine learning algorithms can be used as an additional tool in neuroethology to complement inferential statistics.

Please find below more detailed comments (page numbers are the numbers provided on the reviewer version of the PDF, they might be off by 1 in the authors' version):

Main comments:

P3 L33-34: Authors state that statistical tests are not applicable to raw image data. I don't think this is accurate as there are examples in the literature where pixel values of pictures are concatenated into vectors and multiple images stacked into matrices for multivariate analysis, analyses of similarity, etc. These tests are however performing a different function: comparing observed distributions to expected ones, while ANN is typically used for classification tasks.

Section 3.2: To correct for shielding effects, average pixel intensities are typically calculated in the same manner for similar volumes (or areas) of both the region of interest (neuropils) and of

background signal at the depth of each region of interest. This normalizes for differences between preparations, exposures, etc. Were such steps applied here? If yes, please specify the details in this section.

The authors have employed Tukey's posthoc test to draw comparisons between the three categorical variables. Please clarify if GraphPad Prism 9 corrects for these multiple comparisons by default. If not, employing approaches such as the 'Bonferroni Correction' would be appropriate.

P4 L16 "A few preparations" please specify how many.

Section 3.3.2. Please add references to examples from the literature where this method of data augmentation has been applied. Or to the Keras website given that they describe this in their guidelines.

4.1. If head and brain sizes are correlated, using one of those measures as a covariate in the analysis will help compare differences in the sizes of optic lobe neuropils between active, hibernating, and outside ants while discounting for size-specific effects. Did the variations in sizes of the optic lobe neuropils scale linearly with variations in head or brain size? If not, please clarify why ANOVA was used instead of non-linear models. In addition, have you tried normalizing by the head width? In doing so, are optic lobe neuropil areas over brain width different between H, A, and O individuals?

4.2. When describing significant differences, please add statistical details in the text as well (unless this is against the editorial guidelines).

4.3. Can you quantify the coincidence of the most relevant learning features and the most prominent SI profiles? (cross-correlation?)

P6 L 16: "However, in contrast to our fluorescence measurements (Fig. 3c), the artificial neuronal network discovered significant differences between the lobulae of hibernating *Lasius niger* and both active and outside cohorts (Fig. 6c)." It seems that Fig 4c would be a better reference in the first half of the sentence. Also, and although micrographs shown in Fig. 3 are most likely randomly selected preparations, they show quite a striking difference between the lobulae of Active (3b) and Hibernating (3c) workers. In this context: 1) Please verify that micrographs are labeled correctly, 2) consider re-ordering as H, A, O, as in the rest of the figures, and 3) are there micrographs that would be representative of the mean serotonergic innervation of each group?

ANN discovered differences between the lobulae of ants across the three categories which the fluorescence measurements did not detect. However, in the first case (ANN), the training of the algorithm benefits from data augmentation, while the inferential statistics might have been limited by the sample size. Could the authors comment if performing a bootstrap on fluorescence measurements would have resulted in a different outcome? Bootstrapping fluorescence intensity measurements could be viewed as analogous to data augmentation in ANN. To clarify, this is not necessarily something that is critical for the manuscript, but because authors put forth the idea of ANN performing "better" at detecting subtle differences, it is important to consider whether the statistics used here could have been limited by the sample size.

Figure 7 / Grad-cam: The immunohistochemistry methods used here label serotonergic neurons. So the micrographs used for training the ANN show the labeled serotonin, synapsin, and nuclei. Is this correct? Or did you train the ANN on images showing only the serotonin immunofluorescence? If the latter, then the overlap between the serotonin image and the class-

activation map is expected, right? Because that is the only information available to the ANN during training. Please comment and clarify.

Minor comments:

P2 L32 *Lasius niger* should be italicized.

P3 L9 (and throughout the manuscript) *Lasius niger* should be italicized.

P3 L20 please replace “insects” with “arthropods” because Araneae are not insects but arachnids.

P3 L48 Please consider removing “prematurely”, as it is assumed that if killed, it is before the animal’s natural death.

P4 L19 Please correct as “neuropil borders and reduced”

P4 L43 Please italicize “*Harpegnathos saltator*”

P6 L27 Please italicize “*Drosophila melanogaster*”

Reviewer: 2

Comments to the Author(s)

In this manuscript, the authors seek to find out if the physiological state of hibernation is represented as changes in serotonin expression in the ant optic lobes. They compare serotonin-immunoreactivity in optic lobes of hibernating versus active laboratory-reared ants as well as wild-caught foragers. Immunoreactivity patterns are compared using traditional statistical analyses as well as machine learning algorithms. The questions considered here are interesting, but the experimental design raises major concerns.

1) The primary concern has to do with the handling of the samples used for immunohistochemistry. According to section 3.1 in the Methods, hibernating workers were dissected, and the brains stored for later processing. It is unclear how these brains were stored and for how long. The colony was transferred to room temperature and the next cohort of workers was dissected after 40 days. Were the brains from cohort H in storage that whole time? Were the brains from cohorts A and O also stored for the same amount of time or were all the groups processed at the same time? Many antibodies react very differently in tissue that is stored over a week versus fresh tissue and how that tissue is stored can significantly impact staining. The storage of cohort H brains could account for the decrease in staining. The way this section reads, this experiment was not properly controlled.

2) Different batches of stained tissue can also have variations in staining. Here serotonin-immunoreactivity was represented as “average relative fluorescent units” but it’s unclear how this data was normalized to account for variations in global intensity. Also, was the entire 3-dimensional structure of each neuropil included in the analysis even though the brain was sectioned?

3) Another concern has to do with rearing the ant colonies in the dark. There is no discussion about the developmental effects of this rearing condition on the ant brain, especially the optic lobes. There are significant differences in brain size and serotonin-immunoreactivity between the wild caught and lab reared ants and it isn’t clear that serotonergic neurons developed as they would in wild animals. Why were the ants not provided with an ecologically relevant light cycle? It would also be more convincing if the authors obtain wild-caught ants during hibernation and compare these to the wild-caught foragers. On the subject of wild-caught ants, the authors mention a bimodal distribution of ant sizes in established colonies. Is this size variation related to

division of labor and if so, presumably cohort O were larger because they were foragers. Were ants in the lab reared colonies able to forage and if so, were cohort A individuals foragers or working within the nest? In other words, were apples compared to apples?

4) The introduction would benefit from more details on what hibernation looks like in this ant species. Are all the ants in a state of torpor or do they move within the colony? Do all ants hibernate at the same time or do some remain somewhat active to perform maintenance tasks? Also more detail on the connection between serotonergic signaling and awake versus sleep states would strengthen the rationale for performing this study.

5) It is difficult to interpret the significance of the artificial neuronal network identifying differences between the lobulae of all the cohorts when traditional statistical methods did not. According to the authors, the most relevant learning features overlapped with "the most prominent SI profiles." Was the overall fluorescence equal but the distribution different?

6) One minor issue is that scientific names should be italicized in the manuscript.

===PREPARING YOUR MANUSCRIPT===

===PREPARING YOUR REVISION IN SCHOLARONE===

Author's Response to Decision Letter for (RSOS-210932.R0)

See Appendix A.

RSOS-210932.R1 (Revision)

Review form: Reviewer 1

Is the manuscript scientifically sound in its present form?

Yes

Are the interpretations and conclusions justified by the results?

Yes

Is the language acceptable?

Yes

Do you have any ethical concerns with this paper?

No

Have you any concerns about statistical analyses in this paper?

No

Recommendation?

Accept as is

Comments to the Author(s)

The authors have improved the manuscript and addressed all comments from reviewers.

The only concern left is related to the lack of normalization to background fluorescence or to fluorescence in another brain area. That being said, the fact that there were no significant differences in SI in the lobula, but only in the medulla and lamina, excludes the possibility of a global, brain-wide variation on SI across seasons. In other words, the conclusions drawn by the authors appear to be supported by the data presented here.

Review form: Reviewer 2

Is the manuscript scientifically sound in its present form?

No

Are the interpretations and conclusions justified by the results?

No

Is the language acceptable?

Yes

Do you have any ethical concerns with this paper?

No

Have you any concerns about statistical analyses in this paper?

No

Recommendation?

Reject

Comments to the Author(s)

The authors have addressed all of my concerns except the major issue that the conclusions are based on uncontrolled quantitative immunohistology. If there are no internal controls such as using a relative ratio (e.g., signal1- background1 vs signal2 - background2), then it would be appropriate to include control tissue with each batch. This could be a sample with a known expression level of serotonin or at least tissue from animals held at the same conditions throughout the study. It is not acceptable to quantitatively compare fluorescence without controls as a variety of factors could affect staining even if the authors the same procedure each time.

Review form: Reviewer 3 (Olena Riabinina)**Is the manuscript scientifically sound in its present form?**

Yes

Are the interpretations and conclusions justified by the results?

Yes

Is the language acceptable?

Yes

Do you have any ethical concerns with this paper?

No

Have you any concerns about statistical analyses in this paper?

No

Recommendation?

Accept with minor revision (please list in comments)

Comments to the Author(s)

The main point of the current manuscript is to illustrate the usefulness of an ANN in classifying biological fluorescence images. Thus, in our view, this paper is mostly about a data analysis methods, and less so about the biological implications of these specific results.

Overall, we believe that the manuscript should be published after suggested corrections. We apologize if we repeat some of the comments raised by the previous 2 reviewers, as we were assessing the manuscript independently and without reading the previous comments.

Major comments:

1. Could the authors please provide a step-by-step recipe for the readers who want to apply the same ANN approach to analyse their data? How was this ANN designed? How were the data fed to it? Could the authors make any code available as Supplement (not via Dryad, but via the current journal) and re-written to be usable by someone else? It would be very useful.
2. We did not understand whether any Regions of Interest were selected in the images or not. We presume they must have been, to analyse the regions of the optic lobes. Could the authors please explain in the Methods how these ROIs were selected for all types of data processing?
3. How did the ANN remain blind to the differences in the sizes of neuropiles, in contrast to the brightness of staining? Indeed, was it blind, or it simply learnt the difference in the sizes of

neuropiles? Or the images were cropped such as to only include stained tissue (Page 12 lines 41-44)?

4. Could you explain in more detail for the reader how serotonin can more active in lamina compared to the medulla? Why does the lamina need/have more serotonin? We think the role of each optic lobe neuropile and its relationship to serotonin could be explained in more detail.

5. Please elaborate on the factors that could be responsible for the differences in head sizes between wild and lab animals. Clearly they don't experience the same environmental conditions, but what other factors could be at play here?

6. Could you please clarify in the Introduction, with supporting references, whether hibernation and sleep are the same thing or not? (we think they are not). If you think they are not, please only refer to hibernation and not sleep.

7. We think a more through Discussion is necessary of the previous reported uses of ANN for similar confocal/fluorescence image classification purposes. Please mention examples of previous works, if such exist.

Minor comments:

Page 12, line 28: Rabbit-anti-serotonin ?

Page 13, line 50 : It is a bit strange that the first referred to Figure is Fig 2. Could you please refer to your Fig 1 before referring to Fig 2, or re-number the figures?

Fig 1 legend: *Lasius niger* should be in Italic.

Fig 1: Please provide the three confocal channels separately, in addition to the merged image, to make the serotonin staining more easily visible.

Fig 3: Medulla and lobula of the active (panel b) look much smaller than those of the hibernating (panel a). Also, lobula of "outside" ants looks the same size as "hibernating" in Fig 3. Finally, the magenta (serotonin) staining of the lobula in "hibernating" ants looks very bright on Fig 3, much brighter than the other 2 groups. Lobula serotonin signal also looks brighter in panel b than in panel c. All these points contradict Figure 2. Could you please find more appropriate illustrative images?

Decision letter (RSOS-210932.R1)

Dear Dr Stern

The Editors assigned to your paper RSOS-210932.R1 "Seasonal variations of serotonin in the visual system of an ant revealed by immunofluorescence and a machine learning approach" have now received comments from reviewers and would like you to revise the paper in accordance with the reviewer comments and any comments from the Editors. Please note this decision does not guarantee eventual acceptance.

We do not generally allow multiple rounds of revision so we urge you to make every effort to fully address all of the comments at this stage. If deemed necessary by the Editors, your

manuscript will be sent back to one or more of the original reviewers for assessment. If the original reviewers are not available, we may invite new reviewers.

Please submit your revised manuscript and required files (see below) no later than 21 days from today's (ie 02-Dec-2021) date. Note: the ScholarOne system will 'lock' if submission of the revision is attempted 21 or more days after the deadline. If you do not think you will be able to meet this deadline please contact the editorial office immediately.

on behalf of Dr Jake Socha (Associate Editor) and Pete Smith (Subject Editor)
openscience@royalsociety.org

Associate Editor Comments to Author (Dr Jake Socha):

Associate Editor: 1

Comments to the Author:

The authors have addressed most of the concerns of the first two reviewers. However, there is still a major issue related to control identified by reviewer 1. In response to that review and to help provide additional insight, I invited a third reviewer. The third reviewer is generally positive and has some additional comments to address. Overall, if it is not possible to meet reviewer 1's comments, my opinion is that if the authors are able to provide an open and honest discussion of this specific concern and the limitations of the study within the Discussion of the manuscript, the manuscript can move forward. Publication would then enable debate within the community.

Reviewer comments to Author:

Reviewer: 2

Comments to the Author(s)

The authors have addressed all of my concerns except the major issue that the conclusions are based on uncontrolled quantitative immunohistology. If there are no internal controls such as using a relative ratio (e.g., $\text{signal1} - \text{background1}$ vs $\text{signal2} - \text{background2}$), then it would be appropriate to include control tissue with each batch. This could be a sample with a known expression level of serotonin or at least tissue from animals held at the same conditions throughout the study. It is not acceptable to quantitatively compare fluorescence without controls as a variety of factors could affect staining even if the authors the same procedure each time.

Reviewer: 1

Comments to the Author(s)

The authors have improved the manuscript and addressed all comments from reviewers.

The only concern left is related to the lack of normalization to background fluorescence or to fluorescence in another brain area. That being said, the fact that there were no significant differences in SI in the lobula, but only in the medulla and lamina, excludes the possibility of a global, brain-wide variation on SI across seasons. In other words, the conclusions drawn by the authors appear to be supported by the data presented here.

Reviewer: 3

Comments to the Author(s)

The main point of the current manuscript is to illustrate the usefulness of an ANN in classifying biological fluorescence images. Thus, in our view, this paper is mostly about a data analysis methods, and less so about the biological implications of these specific results.

Overall, we believe that the manuscript should be published after suggested corrections. We apologize if we repeat some of the comments raised by the previous 2 reviewers, as we were assessing the manuscript independently and without reading the previous comments.

Major comments:

1. Could the authors please provide a step-by-step recipe for the readers who want to apply the same ANN approach to analyse their data? How was this ANN designed? How were the data fed to it? Could the authors make any code available as Supplement (not via Dryad, but via the current journal) and re-written to be usable by someone else? It would be very useful.
2. We did not understand whether any Regions of Interest were selected in the images or not. We presume they must have been, to analyse the regions of the optic lobes. Could the authors please explain in the Methods how these ROIs were selected for all types of data processing?
3. How did the ANN remain blind to the differences in the sizes of neuropiles, in contrast to the brightness of staining? Indeed, was it blind, or it simply learnt the difference in the sizes of neuropiles? Or the images were cropped such as to only include stained tissue (Page 12 lines 41-44)?
4. Could you explain in more detail for the reader how serotonin can more active in lamina compared to the medulla? Why does the lamina need/have more serotonin? We think the role of each optic lobe neuropile and its relationship to serotonin could be explained in more detail.
5. Please elaborate on the factors that could be responsible for the differences in head sizes between wild and lab animals. Clearly they don't experience the same environmental conditions, but what other factors could be at play here?
6. Could you please clarify in the Introduction, with supporting references, whether hibernation and sleep are the same thing or not? (we think they are not). If you think they are not, please only refer to hibernation and not sleep.
7. We think a more through Discussion is necessary of the previous reported uses of ANN for similar confocal/fluorescence image classification purposes. Please mention examples of previous works, if such exist.

Minor comments:

Page 12, line 28: Rabbit-anti-serotonin ?

Page 13, line 50 : It is a bit strange that the first referred to Figure is Fig 2. Could you please refer to your Fig 1 before referring to Fig 2, or re-number the figures?

Fig 1 legend: *Lasius niger* should be in Italic.

Fig 1: Please provide the three confocal channels separately, in addition to the merged image, to make the serotonin staining more easily visible.

Fig 3: Medulla and lobula of the active (panel b) look much smaller than those of the hibernating (panel a). Also, lobula of "outside" ants looks the same size as "hibernating" in Fig 3. Finally, the magenta (serotonin) staining of the lobula in "hibernating" ants looks very bright on Fig 3, much brighter than the other 2 groups. Lobula serotonin signal also looks brighter in panel b than in

panel c. All these points contradict Figure 2. Could you please find more appropriate illustrative images?

===PREPARING YOUR MANUSCRIPT===

If you have been asked to revise the written English in your submission as a condition of publication, you must do so, and you are expected to provide evidence that you have received language editing support. The journal would prefer that you use a professional language editing service and provide a certificate of editing, but a signed letter from a colleague who is a fluent speaker of English is acceptable. Note the journal has arranged a number of discounts for authors using professional language editing services (<https://royalsociety.org/journals/authors/benefits/language-editing/>).

===PREPARING YOUR REVISION IN SCHOLARONE===

- 1) One version identifying all the changes that have been made (for instance, in coloured highlight, in bold text, or tracked changes);
 - 2) A 'clean' version of the new manuscript that incorporates the changes made, but does not highlight them.
 - An individual file of each figure (EPS or print-quality PDF preferred [either format should be produced directly from original creation package], or original software format).
 - An editable file of each table (.doc, .docx, .xls, .xlsx, or .csv).
 - An editable file of all figure and table captions.
- Note: you may upload the figure, table, and caption files in a single Zip folder.
- Any electronic supplementary material (ESM).
 - If you are requesting a discretionary waiver for the article processing charge, the waiver form must be included at this step.
 - If you are providing image files for potential cover images, please upload these at this step, and inform the editorial office you have done so. You must hold the copyright to any image provided.
 - A copy of your point-by-point response to referees and Editors. This will expedite the preparation of your proof.

- Ensure that your data access statement meets the requirements at <https://royalsociety.org/journals/authors/author-guidelines/#data>. You should ensure that you cite the dataset in your reference list. If you have deposited data etc in the Dryad repository, please include both the 'For publication' link and 'For review' link at this stage.
- If you are requesting an article processing charge waiver, you must select the relevant waiver option (if requesting a discretionary waiver, the form should have been uploaded at Step 3 'File upload' above).
- If you have uploaded ESM files, please ensure you follow the guidance at <https://royalsociety.org/journals/authors/author-guidelines/#supplementary-material> to include a suitable title and informative caption. An example of appropriate titling and captioning may be found at https://figshare.com/articles/Table_S2_from_Is_there_a_trade-off_between_peak_performance_and_performance_breadth_across_temperatures_for_aerobic_sc_ope_in_teleost_fishes_/3843624.

Author's Response to Decision Letter for (RSOS-210932.R1)

See Appendix B.

Decision letter (RSOS-210932.R2)

Dear Dr Stern,

It is a pleasure to accept your manuscript entitled "Seasonal variations of serotonin in the visual system of an ant revealed by immunofluorescence and a machine learning approach" in its current form for publication in Royal Society Open Science. The comments of the reviewer(s) who reviewed your manuscript are included at the foot of this letter.

on behalf of Dr Jake Socha (Associate Editor) and Pete Smith (Subject Editor)
openscience@royalsociety.org

Associate Editor Comments to Author (Dr Jake Socha):
Associate Editor

Comments to the Author:

Thank you for your revisions, and congratulations on this publication in RSOS!

Appendix A

Dear Dr. Socha,
please find enclosed our revised manuscript RSOS-210932 "Seasonal variations of serotonin in the visual system of an ant revealed by immunofluorescence and a machine learning approach".

We have addressed all the points that you and both reviewers made as follows:

Associate Editor Comments to Author (Dr Jake Socha):

Associate Editor: 1

Comments to the Author:

Both reviewers identified that this study is interesting and provides an original approach. However, there are some major issues that need to be addressed in order for this manuscript to move forward at RSOS. The reviewers have listed numerous concerns. But also please make sure to clarify the main point that machine learning is somehow better than inferential statistics at picking up subtle effects, and address and clarify the potential issue that one treatment group (the hibernation cohort) was kept in a refrigerator for several weeks before immunostaining while the other groups were processed more immediately after dissection.

We thank you and the two anonymous reviewers for the critical and constructive comments that helped to improve our manuscript significantly. We have now clearly stated that we do not assume that machine learning is more powerful than methods of statistical inference but that it is an easier approach to model the kind of image data we have. We also hope that we have resolved the issue of handling and processing of preparations.

Reviewer comments to Author:

Reviewer: 1

Comments to the Author(s)

In the present manuscript, the authors analyzed the seasonal variations in serotonergic signals in the optic lobes of the ant *Lasius niger*. By combining immunofluorescence, brain imaging, and machine learning, the study shows differences in serotonergic innervation between field-caught and lab-reared individuals and suggests seasonal variation in the ants' visual centers. The originality of the manuscript is the use of Artificial Neural Networks to classify hibernating and active ants. The use of Grad-Cam in this context is innovative and the manuscript is well constructed and experiments are carefully designed. The only major comment for the authors is to consider toning down some statements comparing the "performance" of inferential statistics and machine learning algorithms for detecting differences between treatment groups. These two tools are performing different tasks (making inferences using a random sample of data from a given population, versus algorithms that learn how to assign a class label to examples from the problem domain). The manuscript would benefit from revising statements related to this comparison and, instead, highlighting how machine learning algorithms can be used as an additional tool in neuroethology to complement inferential statistics.

We do not assume that machine learning is more powerful than methods of statistical inference but that it is an easier approach to model the kind of image data we have (see below).

Please find below more detailed comments (page numbers are the numbers provided on the

reviewer version of the PDF, they might be off by 1 in the authors' version):

Main comments:

P3 L33-34: Authors state that statistical tests are not applicable to raw image data. I don't think this is accurate as there are examples in the literature where pixel values of pictures are concatenated into vectors and multiple images stacked into matrices for multivariate analysis, analyses of similarity, etc. These tests are however performing a different function: comparing observed distributions to expected ones, while ANN is typically used for classification tasks.

We corrected the statement in the introduction that statistical tests are in principle applicable to image data after changing for example to a vector or matrix representation. We also made more precise how a machine learning model can function as a substitute to a statistical approach. In our case, the idea is that one infers a difference between two sets of images if the machine learning model is able to differentiate the two sets with an accuracy that is better than that of a random classifier.

Section 3.2: To correct for shielding effects, average pixel intensities are typically calculated in the same manner for similar volumes (or areas) of both the region of interest (neuropils) and of background signal at the depth of each region of interest. This normalizes for differences between preparations, exposures, etc. Were such steps applied here? If yes, please specify the details in this section.

Due to the sharp specificity of the serotonin antibody, there is nearly no background signal outside the neuropil (with exception of the cell bodies, which were not in the analyzed sections). We now explicitly write in section 3.2. "...with background in the serotonin channel close to zero and maximum intensities at less than 80% of saturation." We also chose not to normalize to other, non-visual brain areas, because we did not know whether to expect seasonal variations there.

The authors have employed Tukey's posthoc test to draw comparisons between the three categorical variables. Please clarify if GraphPad Prism 9 corrects for these multiple comparisons by default. If not, employing approaches such as the 'Bonferroni Correction' would be appropriate.

Tukey's test as well as the Bonferroni method are designated to control the family-wise error rate (probability of making one or more type I errors) in multiple testing scenarios. While the Bonferroni method attains this goal by correction of the test-wise significance level, Tukey's approach corrects the critical values for the test statistics. Therefore, a combination of both methods is not necessary.

P4 L16 "A few preparations" please specify how many.

We now write that we used two preparations for confocal microscopy.

Section 3.3.2. Please add references to examples from the literature where this method of data augmentation has been applied. Or to the Keras website given that they describe this in their guidelines.

We have now slightly extended this paragraph, giving two references and also referring to the keras website.

4.1. If head and brain sizes are correlated, using one of those measures as a covariate in the analysis will help compare differences in the sizes of optic lobe neuropils between active, hibernating, and outside ants while discounting for size-specific effects. Did the variations in sizes of the optic lobe neuropils scale linearly with variations in head or brain size? If not, please clarify why ANOVA was used instead of non-linear models. In addition, have you tried normalizing by the head width? In doing so, are optic lobe neuropil areas over brain width different between H, A, and O individuals?

Whole brain size was not measured. But we now included head size as a covariate in the ANOVAs, yielding no different conclusions regarding the group effect on the size of lamina, lobula and medulla, respectively. I.e. brain size did not significantly contribute to the models.

We have also now included a plot of neuropil areas versus head width in the electronic supplementary data, which displays significantly linear correlations between neuropils with head width and refer to it in section 4.1. Therefore, no non-linear models are required.

4.2. When describing significant differences, please add statistical details in the text as well (unless this is against the editorial guidelines).

As suggested, we added statistical details in the text (although it makes it harder to read now).

4.3. Can you quantify the coincidence of the most relevant learning features and the most prominent SI profiles? (cross-correlation?)

We are afraid this is a purely qualitative feature. See response to the question about Figure 7, below.

P6 L 16: "However, in contrast to our fluorescence measurements (Fig. 3c), the artificial neuronal network discovered significant differences between the lobulae of hibernating *Lasius niger* and both active and outside cohorts (Fig. 6c)." It seems that Fig 4c would be a better reference in the first half of the sentence. Also, and although micrographs shown in Fig. 3 are most likely randomly selected preparations, they show quite a striking difference between the lobulae of Active (3b) and Hibernating (3c) workers. In this context: 1) Please verify that micrographs are labeled correctly, 2) consider re-ordering as H, A, O, as in the rest of the figures, and 3) are there micrographs that would be representative of the mean serotonergic innervation of each group?

We have corrected the reference to Fig 4c. Regarding the image choice: We had indeed chosen some rather extreme (but beautiful) examples, but have now replaced both the H and A images by examples which more closely represent the mean of those groups. We also have reorganized the figure and ordered it H, A, O according to the rest of the figures. Labeling is correct.

ANN discovered differences between the lobulae of ants across the three categories which the fluorescence measurements did not detect. However, in the first case (ANN), the training of the algorithm benefits from data augmentation, while the inferential statistics might have been limited by the sample size. Could the authors comment if performing a bootstrap on fluorescence measurements would have resulted in a different outcome? Bootstrapping fluorescence intensity measurements could be viewed as analogous to data augmentation in ANN. To clarify, this is not

necessarily something that is critical for the manuscript, but because authors put forth the idea of ANN performing “better” at detecting subtle differences, it is important to consider whether the statistics used here could have been limited by the sample size.

The reviewer makes an interesting point that is frequently associated with data augmentation in machine learning. But data augmentation in machine learning is usually not be regarded as an artificial expansion of sample size to obtain a higher statistical power to reject the null hypothesis. The idea is rather to regularize a machine learning model and to prevent overfitting. As a result, models can reach a higher classification accuracy. In contrast, bootstrapping is used to assess the distribution of the sampling distribution and can be used if distributional assumptions about the data are in doubt. Thus, bootstrapping also prevents overfitting but will also not have a dramatic effect on the testing power. Our argumentation in the manuscript was not that machine learning is more powerful than methods of statistical inference but – as now corrected in the introduction – that it is an easier approach to model the kind of image data we have. To further clarify this point, we added parts of this answer also to section 3.2.2 about the data augmentation.

Figure 7 / Grad-cam: The immunohistochemistry methods used here label serotonergic neurons. So the micrographs used for training the ANN show the labeled serotonin, synapsin, and nuclei. Is this correct? Or did you train the ANN on images showing only the serotonin immunofluorescence? If the latter, then the overlap between the serotonin image and the class-activation map is expected, right? Because that is the only information available to the ANN during training. Please comment and clarify.

The referee is right; we trained the network only on serotonin images. Since we did not detect any significant differences in overall fluorescence between lobulae of different cohorts (Fig. 4c), we wondered what exactly the network learned (it could have, theoretically focused on some random variations at the margin of the image instead of the serotonergic profiles). Thus, we were happy that the network actually learned the serotonergic profiles. Moreover, the method could help to reveal the locations of small differences in other areas of the brain, where the structure is less clearly defined than in the optic lobe.

An explanation for the inability to measure differences in fluorescence could be that the large diameter neurite profiles may contain only a little more serotonin in active than in hibernating lobulae. Due to their low number, as compared to medulla and lamina, this could not be detected by evaluating average fluorescence intensity (we also tried max intensities but found no difference either). One could probably experiment with contrast enhancement and thresholding before measuring fluorescence to reveal the difference, but that would have been a very biased approach.

Minor comments:

P2 L32 *Lasius niger* should be italicized.

P3 L9 (and throughout the manuscript) *Lasius niger* should be italicized.

All species names are now in italics.

P3 L20 please replace “insects” with “arthropods” because Araneae are not insects but arachnids.

We have corrected the term.

P3 L48 Please consider removing “prematurely”, as it is assumed that if killed, it is before the animal’s natural death.

We have replaced “prematurely killed” by “dead”.

P4 L19 Please correct as “neuropil borders and reduced”

We corrected the typo.

P4 L43 Please italicize “*Harpegnathos saltator*”

All species names are now in italics.

P6 L27 Please italicize “*Drosophila melanogaster*”

All species names are now in italics.

Reviewer: 2

Comments to the Author(s)

In this manuscript, the authors seek to find out if the physiological state of hibernation is represented as changes in serotonin expression in the ant optic lobes. They compare serotonin-immunoreactivity in optic lobes of hibernating versus active laboratory-reared ants as well as wild-caught foragers. Immunoreactivity patterns are compared using traditional statistical analyses as well as machine learning algorithms. The questions considered here are interesting, but the experimental design raises major concerns.

1) The primary concern has to do with the handling of the samples used for immunohistochemistry. According to section 3.1 in the Methods, hibernating workers were dissected, and the brains stored for later processing. It is unclear how these brains were stored and for how long. The colony was transferred to room temperature and the next cohort of workers was dissected after 40 days. Were the brains from cohort H in storage that whole time? Were the brains from cohorts A and O also stored for the same amount of time or were all the groups processed at the same time? Many antibodies react very differently in tissue that is stored over a week versus fresh tissue and how that tissue is stored can significantly impact staining. The storage of cohort H brains could account for the decrease in staining. The way this section reads, this experiment was not properly controlled.

We apologize, if we created the impression that we let a batch of brains for months to rot on a shelf, which could indeed be interpreted from the text. What it meant to say was that we could not dissect all brains of a given cohort on a single day, but we collected them from usually three consecutive days of dissection, stored them in the cold room in PBS containing sodium azide (over the weekend), sectioned them, and afterwards processed them together as a batch of sections side by side. This leads to processing times of 2 weeks (from dissecting the first brain to taking the last photo), including several hours at room temperature (for washing steps). We hope this is now explained more clearly in the methods section. Having used Sigma’s serotonin antibody for many years on various invertebrate preparations and even human cell lines we have found the results of this particular antibody to be very stable with very little variation, with the important factor being immediate and thorough fixation, and no obvious effect of other factors like incubation time and temperature. Nevertheless, we are aware of the risk that (unintended) variations of the experimental procedure can add to biological variations as confounding factors, and we have written into the

methods section, that we have attempted to minimize such confounding experimental factors where we could (e.g. , side by side processing of sections in the same solutions, using antibodies from the same frozen aliquot for different ant cohorts dissected more than a month apart). An alternative way to design the study would have been to activate several hibernating colonies, keep the same amount of colonies in hibernation, and later dissect and process them side by side. However, we think that potential variations between (genetically different) colonies would be not less than potential variations between histological processing batches.

2) Different batches of stained tissue can also have variations in staining. Here serotonin-immunoreactivity was represented as “average relative fluorescent units” but it’s unclear how this data was normalized to account for variations in global intensity. Also, was the entire 3-dimensional structure of each neuropil included in the analysis even though the brain was sectioned?

We did not normalize to global intensity. Outside neuropil areas, serotonin immunofluorescence was zero (due to the chosen exposure settings). Within the neuropil, there appeared to be relatively little variation between sections and different animals (with exception of the optic lobes). In any case, since we were looking for such variations in staining, we do not think that it would have been a good idea to normalize against global intensity (there could also be global changes). Of course, one can always argue that variations in staining are not (only) caused by variations in content of the stained antigen, but on variations of the experimental procedure. As explained above, we took care to minimize effects of the experimental procedure by using antibodies, sera etc. from the same aliquots, which we now state explicitly in the methods (section 3.2). As mentioned above, using Sigma’s serotonin antibody for two decades now, I find the staining quality quite stable unless there is a fixation problem.

With regard of the entire 3-dimensional structure: in comparison to the small size of the optic lobe of *Lasius niger*, 60µm sections are actually quite thick. They usually contain the whole lobula, most of the medulla, and a considerable part of the lamina. Nevertheless, if some sections did not contain the entire 3-dimensional structure, in particular of the outer neuropils, an effect on the immunofluorescence measurement would have been minimal. The reason is the highly repetitive columnar structure of the optic lobe neuropils, where each column processes information from a single ommatidium of the compound eye and its surroundings, and is separately innervated by serotonergic profiles (Meinertzhagen and Pyza, 1999) We now include this sentence and reference in the introduction. Serotonin immunoreactivity appears to be evenly distributed over the optic neuropils in the dorso-ventral and antero-posterior axes (Punzo et al., 1994, Hoyer et al., 2005), resulting in frontal sections depicting serotonin distribution in a representative manner for the whole neuropil (as opposed to sagittal sections, which would be problematic). Even if conventional fluorescence microscopy is not as sharply focused as, for instance, confocal laser scanning microscopy, photomicrographs still cover only a fraction of the full thickness of the section, representative for the whole neuropil. We now explicitly write in section 3.2, that photos were taken with a focus on the centre of the neuropil.

3) Another concern has to do with rearing the ant colonies in the dark. There is no discussion about the developmental effects of this rearing condition on the ant brain, especially the optic lobes. There are significant differences in brain size and serotonin-immunoreactivity between the wild caught and lab reared ants and it isn’t clear that serotonergic neurons developed as they would in wild animals. Why were the ants not provided with an ecologically relevant light cycle? It would also be more convincing if the authors obtain wild-caught ants during hibernation and compare these to the wild-caught foragers. On the subject of wild-caught ants, the authors mention a bimodal distribution of

ant sizes in established colonies. Is this size variation related to division of labor and if so, presumably cohort O were larger because they were foragers. Were ants in the lab reared colonies able to forage and if so, were cohort A individuals foragers or working within the nest? In other words, were apples compared to apples?

We have tried to limit the number of variable parameters to a minimum. Thus, we used workers from the same laboratory colonies in both hibernating and active cohorts. Both were also relatively young colonies, before the onset of the size differences in older colonies as reported in the literature. Keeping ants in glass vials and providing them with food without the need to forage would make them (involuntarily) all intranidal workers, thus eliminating the need to discriminate between intranidal and foraging animals. We now write this explicitly in the methods (section 3.1). For intranidal workers, the ecologically relevant light cycle is constant darkness. Keeping them in the dark removes any possible direct effect of light on the optic lobe, thus the only variable parameters remaining being temperature and food supply. If keeping ants in glass vials would have an influence on development of serotonergic neurons, both active and hibernating cohorts would have been affected in the same way.

Regarding size: Recent studies show that in *Lasius niger*, worker size is not directly correlated with internal or external labour [Oktrutniak et al., 2020]. We assume that the large differences in worker size are mainly due to differences in the size and thus the age of the colonies, since also in other ant species the size of the colonies correlates with the size of the workers [Tschinkel et al., 1988]. We have now included these sentences and references in section 4.1.

The suggestion to study wild-caught hibernating ants to wild-caught summer ants is reasonable. However, because of the age- and colony size-related worker size differences, one would need to study the same wild colonies over several years. Furthermore, numerous unobserved confounding factors are likely to influence wild colonies. We think that for a quantitative analysis, a controlled but artificial laboratory environment is more favorable. Fully aware of the fact that the laboratory environment is an artificial situation, we chose outside workers as an outgroup. Interestingly, it turned out that with respect to serotonin immunoreactivity in the lamina, active dark-reared laboratory workers clustered with foraging outside animals rather than with hibernating workers.

In summary, we hope to have compared (cold) apples with (warm) apples, but for comparison also included a batch of pears into the analysis.

4) The introduction would benefit from more details on what hibernation looks like in this ant species. Are all the ants in a state of torpor or do they move within the colony? Do all ants hibernate at the same time or do some remain somewhat active to perform maintenance tasks? Also more detail on the connection between serotonergic signaling and awake versus sleep states would strengthen the rationale for performing this study.

We have included 4 more sentences and 6 references on hibernation in various ants, including *Lasius* species. Since there is not much information on the specific behavior in *Lasius niger*, we have added in section 3.1.: "During hibernation, workers were observed to be either inactive, forming clusters near to the wet cotton or slightly active on the edge of the colony, similar to the behavior of *Leptothorax* workers [Heinze et al.]".

We already write in the introduction about the importance of serotonin (and other biogenic amines) for the control of behavior. Although we do not think, hibernation is equal to sleep, we have in addition included a sentence, and reference, about the involvement of serotonin in sleep regulation

in insects. “Serotonin is involved in circadian rhythmicity, and it is important for sleep regulation in both mammals and insects, where serotonin release promotes sleep [Helfrich-Förster 2018]”.

5) It is difficult to interpret the significance of the artificial neuronal network identifying differences between the lobulae of all the cohorts when traditional statistical methods did not. According to the authors, the most relevant learning features overlapped with “the most prominent SI profiles.” Was the overall fluorescence equal but the distribution different?

As now stated more accurately in the introduction (see also answer to reviewer 1), we used machine learning because statistical models can only be applied to image data after changing their representation (e.g. in form of data vectors or matrices) or to features extracted from the images. We think that there would be more loss of information about what is depicted in the images when using statistical models than when using machine learning models. As a consequence, we assume that the machine learning model – if accuracy is significantly larger than a random classifier – can identify differences where a statistical model can not. Here, indeed, no differences in overall fluorescence were detected in the lobula between cohorts (Fig. 4c).

6) One minor issue is that scientific names should be italicized in the manuscript.
All species names are now in italics.

We would like to thank you and both reviewers for the critical, but helpful comments. We believe that by attending to the reviewers suggestions, the manuscript has considerably improved, and we hope it is now ready for publication in Royal Society Open Science.

On behalf of all authors,

Best regards, Michael Stern

Appendix B

Stiftung Tierärztliche Hochschule Hannover
University of Veterinary Medicine Hannover

Stiftung Tierärztliche Hochschule Hannover, Institut für Physiologie und Zellbiologie
AG Zellbiologie, Bischofsholer Damm 15/102, 30173 Hannover

- the editors -

**Institut für Physiologie und Zellbiologie
AG Zellbiologie**

PD Dr. Michael Stern
Bischofsholer Damm 15/102
(im Physiologischen Institut)
30173 Hannover

Tel. +49 511 856-7767
Fax +49 511 856-7687
michael.stern@tiho-hannover.de

Datum
Hannover, 16.12.2021

ref revision RSOS-210932.R2

Dear Dr. Socha,
please find enclosed our revised manuscript RSOS-210932.R2 "Seasonal variations of serotonin in the visual system of an ant revealed by immunofluorescence and a machine learning approach".

We have addressed all the points that you and all three reviewers made as follows:

Associate Editor Comments to Author (Dr Jake Socha):

Associate Editor: 1

Comments to the Author:

The authors have addressed most of the concerns of the first two reviewers. However, there is still a major issue related to control identified by reviewer 1. In response to that review and to help provide additional insight, I invited a third reviewer. The third reviewer is generally positive and has some additional comments to address. Overall, if it is not possible to meet reviewer 1's comments, my opinion is that if the authors are able to provide an open and honest discussion of this specific concern and the limitations of the study within the Discussion of the manuscript, the manuscript can move forward. Publication would then enable debate within the community.

Thank you for the encouraging suggestion. We still think, that the reviewers' concerns have been addressed, in particular, since reviewer 1 explicitly writes "That being said, the fact that there were no significant differences in SI in the lobula, but only in the medulla and lamina,

Seiten insgesamt
1 / 6

Stiftung Tierärztliche Hochschule Hannover
Institut für Tierökologie und Zellbiologie
AG Zellbiologie (im Physiologischen Institut)
Bischofsholer Damm 15/102
30173 Hannover
Steuer-Nr. 25/202/26506
Ust-ID-Nr. DE 233060166

Bankverbindung
Norddeutsche Landesbank Hannover
BLZ 250 500 00
Konto 106 031 321
IBAN DE30 2505 0000 0106 0313 21
SWIFT-BIC: NOLA DE 2H

www.tiho-hannover.de

excludes the possibility of a global, brain-wide variation on SI across seasons. In other words, the conclusions drawn by the authors appear to be supported by the data presented here”.

Nevertheless, we follow your suggestion and now add a paragraph (4.5. Methodological aspects) to the end of the discussion, addressing possible limitations of the study, and in particular the aspect of unintended variations in the immunolabeling procedure.

Reviewer comments to Author:

Reviewer: 2

Comments to the Author(s)

The authors have addressed all of my concerns except the major issue that the conclusions are based on uncontrolled quantitative immunohistology. If there are no internal controls such as using a relative ratio (e.g., signal1- background1 vs signal2 - background2), then it would be appropriate to include control tissue with each batch. This could be a sample with a known expression level of serotonin or at least tissue from animals held at the same conditions throughout the study. It is not acceptable to quantitatively compare fluorescence without controls as a variety of factors could affect staining even if the authors the same procedure each time.

We again thank the reviewer for the thorough discussion of points of concern. We have already explained that the correctly applied serotonin antibody, because of its sharp specificity, produces no significant background, and thus, subtracting zero from each signal would not lead to any improvement. If there would have been gross global variations between processed batches of sections, this should have affected all neuropils in a similar manner, and could not explain differential effects for instance, on lamina and lobula (as also reviewer 1 has acknowledged, see below).

Reviewer: 1

Comments to the Author(s)

The authors have improved the manuscript and addressed all comments from reviewers.

The only concern left is related to the lack of normalization to background fluorescence or to fluorescence in another brain area. That being said, the fact that there were no significant differences in SI in the lobula, but only in the medulla and lamina, excludes the possibility of a global, brain-wide variation on SI across seasons. In other words, the conclusions drawn by the

authors appear to be supported by the data presented here.

We again thank the reviewer for the thorough discussion and for clarifying their opinion on the unlikeliness of global brain-wide variations of immunolabeling across seasons.

Reviewer: 3

Comments to the Author(s)

The main point of the current manuscript is to illustrate the usefulness of an ANN in classifying biological fluorescence images. Thus, in our view, this paper is mostly about a data analysis methods, and less so about the biological implications of these specific results.

Overall, we believe that the manuscript should be published after suggested corrections. We apologize if we repeat some of the comments raised by the previous 2 reviewers, as we were assessing the manuscript independently and without reading the previous comments.

We thank the reviewer for the helpful comments and suggestions, which we have addressed as follows:

Major comments:

1. Could the authors please provide a step-by-step recipe for the readers who want to apply the same ANN approach to analyse their data? How was this ANN designed? How were the data fed to it? Could the authors make any code available as Supplement (not via Dryad, but via the current journal) and re-written to be usable by someone else? It would be very useful.

We had originally put all this information into the supplementary information, but had, on request of the publisher, moved it into a Dryad repository for the revision. We have now additionally added them back into the supplementary data, but also leave them in the Dryad repository, in the hope that we can satisfy both reviewers and publisher.

2. We did not understand whether any Regions of Interest were selected in the images or not. We presume they must have been, to analyse the regions of the optic lobes. Could the authors please explain in the Methods how these ROIs were selected for all types of data processing?

The regions of interest were the full extent of the neuropils in the section, as delimited by expression of the neuropil marker, synapsin. We now explicitly write this in section 3.2, last paragraph.

3. How did the ANN remain blind to the differences in the sizes of neuropiles, in contrast to the brightness of staining? Indeed, was it blind, or it simply learnt the difference in the sizes of neuropiles? Or the images were cropped such as to only include stained tissue (Page 12 lines 41-44)?

The reviewer's assumption about cropping is correct. We hoped that the indicated sentences did already sufficiently explain this "and reduced to a common resolution (lamina: 100x500 pixels, medulla: 400x800 pixels and lobula: 400x250 pixels)". In addition, we now explicitly write that we used the outline of the synapsin immunoreactivity as a neuropil marker to determine the borders of the neuropil area.

4. Could you explain in more detail for the reader how serotonin can more active in lamina compared to the medulla? Why does the lamina need/have more serotonin? We think the role of each optic lobe neuropile and its relationship to serotonin could be explained in more detail.

We have introduced three sentences and two new references about information processing and serotonergic modulation in the optic lobe neuropils in section 4.4: "In the insect optic lobe, visual information is processed by columnar interneurons with increasing receptive field size along the hierarchically ordered neuropil areas lamina, medulla, and lobula, in each of which distinct features like colour, shape, or movement are processed by local circuits and relayed to deeper brain centres by tangential neurons [Douglass and Strausfeld, 2003]. These information processing circuits in the insect optic lobe are under neuromodulatory control [Cheng&Frye 2020]. 5HT is known to modulate neuronal responses of neurons in all three visual ganglia (lamina, medulla, lobula), but since in most studies serotonin had been bath-applied, the exact site of 5HT action is not clear. Indirect information about the sites of 5HT action can be derived from the distribution of serotonin receptors in all three optic lobe neuropils [67]."

In the same paragraph, we have also included a sentence speculating about the significance of the conspicuous differences between active and hibernating ants in the lamina (importance of circadian activity regulation for actively foraging as opposed to hibernating colonies).

5. Please elaborate on the factors that could be responsible for the differences in head sizes between wild and lab animals. Clearly they don't experience the same environmental conditions, but what other factors could be at play here?

We have already explained this in detail, including three references, in section 4.1.: "Older and larger *Lasius niger* laboratory colonies tend to produce workers of different sizes with a two-peak size distribution [11]. Recent studies show that in *Lasius niger*, worker size is not directly correlated with internal or external labour [61]. We assume that the large differences in worker size are mainly due to differences in the size and thus the age of the colonies, since in other ant

species the size of the colonies correlates with the size of the workers [62]. Since there is a correlation between head and brain size in ants [26], such variations in population size would influence measurements. In our young colonies, we can exclude size variation as a confounding factor.”

We can think of no other explanations.

6. Could you please clarify in the Introduction, with supporting references, whether hibernation and sleep are the same thing or not? (we think they are not). If you think they are not, please only refer to hibernation and not sleep.

We do not think hibernation is equal to sleep, and did not write about sleep in the original manuscript, but had to include a reference to sleep on explicit request of reviewer 2. We hope that we can keep the sentence without confusing the readers too much.

7. We think a more thorough Discussion is necessary of the previous reported uses of ANN for similar confocal/fluorescence image classification purposes. Please mention examples of previous works, if such exist.

To the best of our knowledge, no similar reported uses of ANNs to detect differences in insect brains exist, but we have included three further references of studies involving machine learning approaches on confocal images in pathology and toxicology in the introduction.

Minor comments:

Page 12, line 28: Rabbit-anti-serotonin ?

We corrected the mistake.

Page 13, line 50 : It is a bit strange that the first referred to Figure is Fig 2. Could you please refer to your Fig 1 before referring to Fig 2, or re-number the figures?

We now refer to Fig. 1 in the methods section 3.2.

Fig 1 legend: *Lasius niger* should be in Italic.

In the Figure captions section (p. 17, l. 46) *Lasius niger* is correctly in italics, but for the captions separately added to the uploaded figures, we were unable to choose fonts, styles, etc. in the online submission system.

Fig 1: Please provide the three confocal channels separately, in addition to the merged image, to make the serotonin staining more easily visible.

As suggested, we now provide the single channels separately as well (Fig. 1c-e).

Fig 3: Medulla and lobula of the active (panel b) look much smaller than those of the hibernating (panel a). Also, lobula of “outside” ants looks the same size as “hibernating” in Fig 3. Finally, the magenta (serotonin) staining of the lobula in “hibernating” ants looks very bright on Fig 3, much brighter than the other 2 groups. Lobula serotonin signal also looks brighter in panel b than in panel c. All these points contradict Figure 2. Could you please find more appropriate illustrative images?

We had chosen better images in the original manuscript, but replaced them on request by reviewer 1. We hope that the version we present now is a little more representative for the quantitative data. Nevertheless, one has to keep in mind that hardly any image exactly matches the average (that’s the point of averaging). The images should give the reader a general impression on what serotonin-immunoreactivity looks like in the ant optic lobe.

We would like to thank you and all three reviewers for the critical, but helpful comments. We believe that by attending to the reviewers suggestions, the manuscript has considerably improved, and we hope it is now ready for publication in Royal Society Open Science.

On behalf of all authors,

Yours sincerely

Michael Stern